# Psychosocial Considerations for the Child with Rare Disease: A Review with Recommendations and Calls to Action

**DOI:** 10.3390/children9070933

**Published:** 2022-06-21

**Authors:** Leslee T. Belzer, S. Margaret Wright, Emily J. Goodwin, Mehar N. Singh, Brian S. Carter

**Affiliations:** 1Division of Developmental and Behavioral Health, Section of Pediatric Psychology, Children’s Mercy Kansas City, Kansas City, MO 64108, USA; 2Department of Pediatrics, School of Medicine, University of Missouri-Kansas City, Kansas City, MO 64108, USA; smwright@cmh.edu (S.M.W.); ejgoodwin@cmh.edu (E.J.G.); bscarter@cmh.edu (B.S.C.); 3Division of General Academic Pediatrics, The Beacon Program, Children’s Mercy Kansas City, Kansas City, MO 64111, USA; 4School of Medicine, University of Kansas, Kansas City, KS 66160, USA; 5Department of Psychology, Clinical Child Psychology Program, University of Kansas, Lawrence, KS 66045, USA; msingh4@cmh.edu; 6Department of Medical Humanities & Bioethics, University of Missouri-Kansas City, Kansas City, MO 64108, USA; 7Bioethics Center, Children’s Mercy Kansas City, Kansas City, MO 64108, USA

**Keywords:** rare disease, children, families, medical complexity, care coordination, psychosocial, policy, advocacy

## Abstract

Rare diseases (RD) affect children, adolescents, and their families infrequently, but with a significant impact. The diagnostic odyssey undertaken as part of having a child with RD is immense and carries with it practical, emotional, relational, and contextual issues that are not well understood. Children with RD often have chronic and complex medical conditions requiring a complicated milieu of care by numerous clinical caregivers. They may feel isolated and may feel stigmas in settings of education, employment, and the workplace, or a lack a social support or understanding. Some parents report facing similar loneliness amidst a veritable medicalization of their homes and family lives. We searched the literature on psychosocial considerations for children with rare diseases in PubMed and Google Scholar in English until 15 April 2022, excluding publications unavailable in full text. The results examine RD and their psychosocial ramifications for children, families, and the healthcare system. The domains of the home, school, community, and medical care are addressed, as are the implications of RD management as children transition to adulthood. Matters of relevant healthcare, public policies, and more sophisticated translational research that addresses the intersectionality of identities among RD are proposed. Recommendations for interventions and supportive care in the aforementioned domains are provided while emphasizing calls to action for families, clinicians, investigators, and advocacy agents as we work toward establishing evidence-based care for children with RD.

## 1. Introduction

Rare diseases (RD) affect children, adolescents, and their families infrequently, but have a significant impact. While the US National Institutes of Health defines such conditions as occurring with a frequency of 1 in 200,000 or less, definitions vary around the world [1]. The US definition is linked to what is known as the 1983 Orphan Drug Act—a measure taken to provide incentives to the pharmaceutical industry to research and develop new treatments for infrequently occurring conditions [2]. Another name for these conditions in the US is “orphan diseases” as the law was called the Orphan Drug Act (www.eurordis.org/about-rare-diseases accessed on 15 June 2022).

In the European Union (EU), RDs have been defined as those occurring in less than 1 per 2000 people [3]. Like the US orphan drug laws, attention to orphan diseases and pharmaceuticals exists in the EU and Japan [4]. However, the absence of governmental attention to RDs and their impact continues to be heralded worldwide [5]. This is certainly understandable as 40 per 100,000 individuals have RD [1]. In such a calculus, it is likely that around the world there are hundreds of millions of people affected by RD with estimates of 25 million people or more in North America and up to 36 million in Europe [1]. Genetic disorders account for a large majority of RD [6]. While identifying a genetic basis for RD may allow refined definitions and approaches in research and development for therapeutics, it also makes the recognition and attention to certain disease features difficult due to the specificity of gene-mediated processes. Features may entail a phenotypic expression that is recognizable to the clinician, such as seizures, yet be minimally responsive to typical therapeutic agents and require symptom-driven genomic investigation or specific tests. In the case of RD presenting as epilepsy, this may involve special testing of the cerebrospinal fluid. Other conditions may involve metabolic processes impacting vital organ function, or significant developmental behavioral differences [7]. The diagnostic odyssey that must be undertaken as part of having a child with RD is immense and carries with it practical, emotional, relational, and contextual issues that are not well understood by those caring for or impacted by the child’s RD.

Children with RD often have chronic, complex medical conditions requiring a complicated milieu of care by numerous clinical caregivers. They may feel isolated, have anxiety or depression, and may feel stigmas in settings of education, employment and the workplace, lacking self-sufficiency, and feel a lack a social support or understanding by others [8]. Families of children with RD face a multitude of challenges, too. Parents of children with neurodevelopmental differences have expressed feelings of social isolation and being overwhelmed [9]. Other parents report not being understood by their peers with more typically developing children and facing a veritable medicalization of their homes and family lives [10,11,12,13]. 

This review examines RDs and their psychosocial ramifications for children, families, and the healthcare team. The domains of the home, school, community, and healthcare are addressed, as are the implications of RD management as children transition to adulthood while families must access and advocate for their healthcare needs. To this end, matters of relevant healthcare and public policy and more sophisticated translational research that address the intersectionality of identities among RD are proposed. Finally, recommendations for evidence-based interventions and supportive care in the aforementioned domains are presented. Calls to action for families, clinicians, investigators, and advocacy agents are stated to prompt continued work toward establishing evidence-based care for these children. See Figure 1.

## 2. Materials and Methods

The initial search of the literature was conducted in PubMed and Google Scholar databases for peer-reviewed articles published until 15 April 2022, in English. The secondary search included articles from reference lists that were identified in the primary search. Records were screened initially by title and abstract and then full-text articles were retrieved for full review and eligibility evaluation. The searches combined a range of key terms including “Pediatric” OR “Children” AND “Rare Disease” AND “Psychosocial” OR “Social.” Duplicate manuscripts and those unable to be obtained by interlibrary loan or through library access online were removed after exporting references to Endnote Online (https://www.myendnoteweb.com/EndNoteWeb.html, accessed on 1 February 2022). The reader is referred to this Special Issue (https://www.mdpi.com/journal/children/special_issues/psychosocial_considerations accessed on 10 June 2022) for research on the following relevant topics that were published outside of the time range for inclusion in the current review article, e.g., psychosocial difficulties among preschoolers, advance care planning, social support for siblings of children with cancer, transition to adulthood among youth with RD, and experiences in youth with rare cancers.

## 3. Results

### 3.1. The Child with RD

Individuals with RD are surviving and living lives not previously thought possible, yet are not well understood [15]. Existing knowledge about children with RD describes the psychosocial experiences of their families and caregivers [16,17] yet the child with RD must remain the focal point. Children with RD are more likely to encounter significant challenges in their functioning at home, at school, and in their community [18]. However, there is scant research characterizing the child’s own experiences. What is known about RD is drawn from the intersection of groups of children with chronic disease. For example, children (and youth) with special healthcare needs (CSHCN or CYSHCN, used interchangeably in this review) have, or are at increased risk of having, a chronic physical, developmental, or emotional condition requiring more healthcare services than is needed by most typically developed children [19]. Children with medical complexity (CMC) are a sub-population of CSHCN and children with RD, often with functional impairment and dependence on medical technology and equipment (such as gastrostomy tubes).

The medical and psychosocial needs of CMC and their families are not well met by many existing healthcare models, and CMC are more than twice as likely as typically developing children to have unmet healthcare needs [20]. They represent a small portion of children but account for more than one third of pediatric healthcare resources consumed annually [21]. Children with RD require multidisciplinary care coordination among multiple sectors of care [22,23,24], experience a higher frequency of inpatient stays than others, and experience significant obstacles to having their voices heard. Studies have shown variable efforts for care teams to involve the child with an RD in their own care, relying on parent caregiver reports predominantly [25]. While not all CSHCN and CMC have RD, many do, and the intensity and acuity of their medical and psychosocial needs make this a valuable group to consider when assessing the psychosocial needs of children with RD.

#### 3.1.1. Intersecting Identities and Experiences

The intersecting facets of the child’s identity and experiences are ignored in studies about RD. Understandably, studies about RD may be impacted by the small number of individuals with a rare diagnosis. Yet, without attention to intersectionality and understanding our care models and for whom and under what conditions they are effective, the insights gained can be limited in their utility and application [26]. Social determinants of health (SDH) are known to contribute to inequity in outcomes, although the impact on children with RD has not been fully characterized. Diagnostic genetic testing can be a powerful tool, although it may not be available to all in need [27]. In the USA, CSHCN are disproportionately of Black, non-Hispanic/Latinx heritage and are believed to experience inequities in health and healthcare access due to historical marginalization [28]. When inequities are examined based upon race and ethnicity, primary language spoken in the household, insurance type, and poverty status, children with medical complexity are found to be twice as likely to have at least one unmet need, compared to children without medical complexity. However, in one study, children with medical complexity had disproportionately higher unmet needs than children without medical complexity across all categories of race and ethnicity [20]. Because racial, geographic, and socioeconomic inequities impact healthcare globally, these SDH are estimated to have substantial impact on children with RD. 

#### 3.1.2. Behavioral Health

Health-related quality of life (HRQOL) in children with chronic disease is known to be related to both self-management and self-efficacy [29]. Children living with RD experience barriers that impact their quality of life (QOL) and psychosocial functioning [30] as demonstrated by higher levels of mental health needs [31]. In a recent cross-sectional study conducted in Western Australia, 43.9 percent of parents of children with RD reported that their child experienced mental health difficulties [32]. Lum and colleagues found that parents of children with chronic illnesses were 2.2 times more likely to report that their child experienced emotional distress and lower levels of self-confidence [33]. 

#### 3.1.3. Communication

Understandably, children with RD that have emotional distress or comorbid developmental behavioral conditions are at risk for having communication challenges. These children may experience communication barriers, some negatively impacting their care, or they may have a comorbid speech or language disorder. The psychosocial impacts of having a speech or language disorder have been documented and include bullying, delays in adaptive functioning, and difficulties with emotion regulation [34,35]. Others noted that children with language disorders are more likely to experience anxiety, depression, ADHD, and externalizing behaviors compared to those without language disorders [36]. In examining psychosocial outcomes, Lewis and colleagues found that adolescents with early childhood speech sound disorder experienced poorer psychosocial outcomes when combined with language impairment [37]. Thus, children with RD coupled with communication disorders may require a different type of support in promoting positive psychosocial outcomes.

### 3.2. RD in the Family

RDs impact the entire family. Recently, Hoover and colleagues poignantly acknowledged that the ongoing COVID-19 pandemic has brought new visibility to difficult experiences that are commonplace among families of CSHCN [38]. Examples of such a family impact include being forced into homeschooling, being homebound, stretching oneself to meet the social and educational needs of the child while meeting their health needs, inequities in quality of healthcare, and the injustices of their outcomes linked to race, ethnicity, or socioeconomic status. The authors make clear that this previously invisible role of family caregiving—including the work of nurturing, tasks, resources and services to meet day-to-day needs—is not yet adequately recognized [38]. In fact, despite challenges they face, families must consistently respond to both routine day-into-night cares as well as the crises that arise [13] regardless of whether prior knowledge or support services are available to help. This is especially important to acknowledge given families’ resilience in the face of adversity.

### 3.3. Social Determinants of Health

SDHs have a significant impact on children with RD and their families. Families of children with complex chronic conditions, including those with RD, are more likely to experience medical financial hardship [39,40]. Medical financial hardship is correlated with negative child health outcomes regardless of a family’s socioeconomic status or other financial resources [41]. There is a strong association between foregone family employment [42] and family-provided medical care [43,44,45]. CSHCN have elevated risk for food insecurity and malnutrition, which has a dramatic impact upon daily and long-term functioning [46,47]. Reduced access to household materials is associated with increased acute healthcare utilization and unmet healthcare needs among CSHCN [48,49]. Beyond housing stability, this also relates to accessible housing adaptations for children with disabilities. Families face difficult trade-offs when it comes to identifying viable housing options [50]. High acuity medical episodes, such as prolonged intensive care hospitalization or development of new medical technology dependence may bring heightened vulnerability [51]. For example, the proportion of families with unmet basic needs increases during chemotherapy treatment for newly diagnosed pediatric cancers [52]; the same may be true for acute changes in health status with other conditions such as RD.

### 3.4. Home Care

Some children with RD have complex medical needs that require chronic home health services. Private duty nursing (PDN), also called “home nursing care,” is an integral part of care for some children [53]. Examples of PDN services include tracheostomy/ventilator and other airway and pulmonary care, providing tube feedings, administration of medications, performance of ordered home therapy exercises, and other essential health services. Without PDN, some children may not live safely at home. Limited access to home healthcare services and staffing in North America, Europe, and globally has been identified as a crisis for families. PDN is known to be inextricably linked with child survival and family stability due to family life being intertwined with home healthcare schedules, staffing, and services [54]. In fact, gaps in PDN staffing threaten family physical, mental, and financial wellbeing. Families must continually fight payors and government agencies for their allotted services, with out-of-pocket costs being customary although unjust. This long-standing widespread shortage of home nurses and geographic heterogeneity of both quality and quantity of nursing services mean that many children do not receive the number of hours for which they qualify [43,55]. This results in family caregivers improvising nursing care, which has the potential to place the child’s health at risk, can result in parent(s) foregoing employment and income [44], and negatively impact marital and family dynamics [56].

### 3.5. The Search for Answers

Families experience myriad psychosocial challenges in providing loving care to their child with RD. Initially, the diagnostic odyssey of identifying the genetic underpinnings of a rare disease can raise many poignant issues for families [57,58]. Mendelian genetic disorders are primarily caused by alterations in one gene or abnormalities in the genome and may be seen since birth or visible in the family history. Although Mendelian genetic disorders are individually rare, they are collectively more common and contribute disproportionately to pediatric morbidity and mortality [27]. Genetic testing allows for the benefit of individualized treatment plans in addition to ending the diagnostic odyssey, which not only halts further unnecessary testing but may also result in immense psychological benefit, leading to improved quality of life. However, genetic testing may reveal that other family members carry the same gene or disorder, which can be difficult for families to navigate [59]. Furthermore, ensuring equitable application of these advances in genomic technology has been challenging. Technology has limits, too. Even when expanded genome sequencing is available, it may not yield an interpretable answer, or after many years answers obtained may only facilitate a small step toward better understanding or treating RD.

### 3.6. Barriers to Wellbeing

While genetic diagnosis may provide timely medical intervention, informed choices, access to clinical trials and engagement in disease-specific support [59] for some with RD, significant barriers to wellbeing also emerge. This includes isolation and loneliness. At times, in being one of the only or few to receive a specific diagnosis, often with little known about its course, prognosis, or known interventions, loneliness prevails [60]. While in most circumstances outside of RD, a diagnosis brings understanding, treatment and reasonable prognostication, a diagnosis of an RD is accompanied by uncertainty. This can increase anxiety about the future, create instability, and lead to a variety of sequelae for family members. Finding a new normal following this journey may be daunting for many [7]. For example, parents of CMC report lower health-related quality of life (HRQOL) when compared to parents of non-CMC, and ratings of mental health QOL are lower than physical ones [61]. However, families may experience lack of access to suitable mental health services or experience a “lack of fit” in peer support groups [62]. Behavioral health challenges for family caregivers are largely known to include caregiver stress [63,64]. Family caregivers often perceive that they do not have time to address their own behavioral health needs [60,61]. When they do try to attend to their own needs, they may not be able to access appropriate services due to caregiving-specific barriers to care [65,66,67]. 

Cardinali and colleagues noted that challenges reported in caring for a child with RD often varied for mothers and fathers [68]. Both valued information about the diagnosis, perceived the lack of an organized medical system, and shared many feelings and behaviors as a couple. Fathers noted challenges with finances, education, feelings, and behaviors. Mothers noted problems with career, adaptation to the child’s needs, their role in education, their own feelings, and how the family functioned as a system. While finding others who truly relate to the unique aspects of the individual RD is valuable to the family, they may find that others with RD share themes of common experiences [68]. Positive family functioning has been demonstrated to positively influence the QOL for children with RD [69,70]. Family cohesiveness, positive intrafamily relationships, and acceptance are related to positive family and child functioning; in fact, some families have created positive meaning from their experiences [70,71].

Parents describe feeling misunderstood by family and friends regarding the realities of their daily caregiving experience, and many describe difficulty connecting with a supportive community [9]. Social support and respite care are known to sustain caregiver wellbeing [13] and reduce stress and burden [11,72]. However, the logistics of accessing these helpful resources are rife with barriers [73]. Access to informational and interactive peer support for parent caregivers of children with RD is a substantial service to families [13,74]. These may take the form of in-person events, group offerings, virtual live meeting rooms, or asynchronous communication forums such as social media or chat rooms [57]. Despite the accessibility of virtual social support options, parent caregivers may experience significant behavioral health challenges that can be exacerbated by caregiving demands and all that comes with caring for RD [75].

### 3.7. Coordination of Care

Access to coordination of needed services represents a significant challenge. RDs typically require multiple specialists and thus multiple appointments that must be coordinated and attended by the family, many of which may have little experience in complex care settings [60,61,64]. While a patient- and family-centered care (PFCC) approach to children’s chronic conditions is often emphasized [76], care programming has a long way to go to address the needs that arise for families of a child with RD [11]. Often, children with RD are seen in clinics without established care pathways and this can be experienced by the family as an ongoing struggle to advocate for their child’s needs [77]. Additionally, healthcare teams may not be familiar with the RD.

### 3.8. Access to Information

In addition to challenges accessing coordinated care, it can be challenging to access accurate and helpful information pertinent to the RD. Lack of or limited access to accurate evidence-based information about their child’s condition can contribute to additional caregiver stress. Managing the unknown and when contending with situations where there is no answer or information is a challenge that is salient to families of a child with RD. Families search the internet often for answers to questions, with information quickly at their fingertips that may or may not be accurate. The dangers of misinformation are substantial with this approach. Even in cases where information is available about the disease, it may not be available in a caregiver’s primary language, compounding access inequities and stress [78,79]. Families report asking healthcare providers for answers and being told they do not know, or that information is not yet known. While care teams seek the most up to date information in the care of their patients, families may be asked to tolerate ambiguity. 

Caring for children with RD requires adequate personal health literacy including the skills to find, understand and use information to inform health-related decisions. These skills include reading, listening, speaking and numeracy skills as well as the ability to seek and find information from reputable sources [80]. An individual’s personal health literacy can be dynamic and may be lower in times of stress [81]. A child with RD may also have multiple caregivers who are expected to follow complicated instructions relating to medical care such as medications, nutrition, and equipment. 

### 3.9. The Family Is Part of the Care Team

Despite the challenges that families report, the experience they gain over time can be significant. They become ‘expert medical caregivers’ by experience, sometimes about RD, but most certainly about their own child and their care [38]. Parents become to the intermediary between care teams and their child or may even find themselves in a teaching role in explaining their child’s RD to the care team. 

#### 3.9.1. Advisory Councils and Boards

One pathway for care coordination and patient- and family-centered care builds upon the expertise of the family caregiver. Patient and family engagement refers to the process through which these individuals are included in the diagnostic, treatment, and administrative processes. These groups bring patients and families together to provide guidance on how to improve the patient and family experience. Involvement in these councils is one way to ensure patients and family members are engaged with their health-care experience. In fact, many hospitals and healthcare organizations have formed patient and family engagement programs, such as patient family advisory councils (PFAC) or family advisory boards (FAB) that recruit parents of a child with an RD to serve on a patient family advisory council or board.

These parent or patient volunteers serve to provide valuable input to care teams about issues that directly impact patient care and family wellbeing. Through this partnership, parents assist in the expert care of other children served by the healthcare system. Parents may feel heard and healthcare teams may stay connected to the true purpose and concerns of families. Parents or patients engaged in this role may enhance intervention outcomes as told by families in real-time so that iterative improvements may be made to impact their child(ren) and others served in the organization. The challenge, however, is that families serving in these roles may not be representative of the entire population they represent. Additionally, parents of children with rare diseases or medical complexity may not have time to volunteer in this role. See Figure 2. 

#### 3.9.2. The Siblings

Sometimes, the family member that is unseen or in the background, siblings of children with RD, present unique experiences and serve roles in the family that differ from families without RD. Deavin and colleagues conducted a meta-synthesis of multiple qualitative studies to draw conclusions from the direct reports of siblings themselves to better describe the psychosocial commonalities experienced by siblings in order to improve care for families [82]. They identified two overarching themes experienced by siblings: (a) relationship changes and (b) managing change. Within these themes existed the family’s relational changes in cohesion and relationship between parents and the sibling, as well as the sibling’s relationship to self and contending with the emotional experience of foregoing their own needs and serving new roles and responsibilities. 

The Committee on Psychosocial Aspects of Child and Family Health of the American Academy of Pediatrics summarized challenges for siblings as follows [83]. A common role that siblings report is that of the assistant caregiver. Siblings may feel overshadowed or neglected due to the constraints of the child patient’s needs and the impact on limiting parental time and resources toward the sibling. They can feel embarrassed if others stare at or make comments about the family member with RD because they look different. They may become angry if they are asked to assume more household chores or guilty when they resent their added responsibilities in the family. Additionally, siblings report feeling guilty about being healthy and not having RD. Siblings who may be genetic carriers of an RD that is not phenotypically apparent may feel guilt and anxiety about what this means for their own decisions to have families of their own. Additionally, siblings may feel anxious about becoming ill themselves and experience a higher rate of medical trauma related to witnessing intense medical experiences of their sibling at home or within the medical setting. While academically siblings may experience more missed school, some studies report academic challenges that extend beyond missed days due to hospitalizations or medical visits for their sibling [84]. 

### 3.10. Access Barriers in the Community

Children with RD and their families may experience barriers to engagement in community activities due to exclusion or limited access for individuals with physical or intellectual differences. This affects the families’ ability to thrive in the communities where they and their families live, learn, work, and play. 

#### 3.10.1. Community Activities and Transportation

Lack of wheelchair accessible transportation, as well as other adaptations to meet the mobility needs of some children with RD, create significant barriers to attendance at medical appointments, engagement in educational and therapeutic activities, and play [85,86,87,88]. Some legal statutes exist to help ensure accessibility, such as the Americans with Disabilities Act (ADA) in the USA, although in reality these provisions often fall short and can at times require costly and time-intensive enforcement measures if not being followed. Without reliable community options for accessible transportation, families face the choice to either pay to adapt a personal vehicle or limit participation in activities outside the home. The sheer cost and lack of funding mechanisms for these adaptations create an insurmountable barrier for many families, exacerbating social isolation and limiting access to medical and educational services in the community. Families in both rural and urban areas face significant, although at times different, barriers to accessible transportation. 

Social exclusion and isolation are commonly described among parents of children with RD [89]. Parents of children with rare neurodevelopmental disorders have described experiencing social taboo and stigma when interacting with families of neurotypical children [9]. Exclusion can be related to limitations of the built environment or structure of an activity (e.g., playground without accessible equipment) or to assumptions about the nature of a child’s disability (e.g., child excluded from a reading activity due to the assumption they would not comprehend it). Some children experience barriers to support services and special programs due to the rare nature of their qualifying condition [15]. For example, a child with a progressive neurodegenerative disorder caused by a novel genetic mutation may not be technically eligible for a waiver program because their diagnosis is not one of the listed eligible conditions. Eligibility criteria for support programs are heterogenous, the application processes can be onerous, and once accepted the waitlist for services can be years long.

#### 3.10.2. Appropriate and Fair Education

Children with RD have the right to an appropriate education in the least restrictive environment possible, but school districts and service providers are not always equipped with the resources and skills to meet that need. Legal statutes such as the Individuals with Disabilities Education Act (IDEA) in the USA set expectations for school-based services for children with disabilities, but the unfortunate reality is that individual school districts do not always have the necessary resources to fulfill these expectations. Parents of children who have neurocognitive differences, including children with RD, often find themselves having to advocate for their children’s educational and therapy needs. The lack of appropriate support services to allow for appropriate access to places of learning may prevent children with RD from attending school or being educated according to their rights and abilities. 

Children with RD may also experience social difficulties by way of stigma and bullying and face greater misunderstandings of their experiences by peers and teachers [30,32,90] and are more likely to be bullied compared to their peers [91]. Delays in diagnosis may impact school planning and access to resources for children with RD [92,93]. Given that schooling may be disrupted by the child’s medical needs, official recognition of abilities, coordination of care, curricular adaptation, emphasis on autonomy, and peer support may all influence an equitable education [92,93]. Furthermore, children with RD experience disruptions in their school experience and are more likely to have higher academic, medical, and social–emotional needs but do not experience school-based support at the same level [33]. Common themes in a qualitative study where children with RD experienced reduced school attendance included increased discrimination, reduction in participation, and facing students and teachers who lack knowledge and understanding of their experiences [92]. 

### 3.11. The Healthcare Team

Caring for a child with RD is a ‘team sport’ and ‘takes a village.’ Due to frequent healthcare utilization and fragmentation of the healthcare system, individuals with RD and their families often must update new providers about their child’s complex history and care needs, feeling more like an expert than the professional, telling the story over and over. Turnover of health professionals has also been identified as a concern for those with RD. Families have identified the need to continually update existing or new “continuity” providers as a stressor and dissatisfier. Healthcare organizations and healthcare teams can optimize care coordination in multiple ways. Practices should work to minimize wait times for acute and chronic care and maximizing access to services. Children with RD and their families may rely on a multidisciplinary coordinated care team in that children with chronic conditions often are higher utilizers of social work services than their healthy counterparts [73].

Healthcare teams not only consist of many professionals, but also trainees. Trainees need to be made aware of unique needs of individuals with RD. It is important that clinicians model principles of patient- and family-centered care to their trainees. Teams should have shared goals, clear roles, mutual trust at effective communication as well as measurable outcomes [94]. Healthcare teams should model and instill these principles in medical trainees and other interprofessional education. Healthcare teams can learn from one another as well as from patients and families. Project ECHO is one model that has proven effective across disciplines and uses a “spoke and hub” model for bidirectional education and case-based learning via tele-mentoring [95]. This type of model moves knowledge instead of people. For healthcare teams, educators or individuals with RD, the ECHO model may serve as a potential way to rapidly share best practices and information [95]. 

Medical Homes are one example of a system that meets patient needs, improves the patient experience, and also improves provider efficiency and support [96]. A Medical Home is a beneficial component of care for all children, especially those with RD, disabilities, or other medical complexity [9]. In the USA, they have demonstrated healthcare cost reductions to families and insurance payors as well as reductions in emergency services utilization [97]. As children with RD have so many healthcare providers, it is sometimes difficult to identify the primary person or team responsible to see the whole picture and coordinate care [13]. The Medical Home provider may be a specialist (such as geneticist or neurologist office that has a care team familiar with the RD) or a primary care provider that is the central ‘home’ for general healthcare and coordination for the child with RD [98]. The healthcare team that serves as the Medical Home may differ at times by health condition and geographic resources; it may also change over time as patient and family needs or resources shift. Centering care within the Medical Home model ensures continuity and minimizes the need for families or individuals with RD to retell their story. The model also enables a big picture view of the health and wellbeing of the whole child with RD, not only one aspect of care, and how different medical recommendations from multiple specialists interact [99]. While some conditions have entire clinics dedicated to individuals with a certain diagnosis (e.g., a Down syndrome clinic, or Neurofibromatosis clinic), individuals with RD will not typically find such clinics for their condition based on its low prevalence. Rather, multidisciplinary complex care clinic programs with interprofessional healthcare teams (dietician, pharmacist, primary provider, social worker, psychologist, nursing, etc.) are an example of care coordination programs for individuals with RD or multiple health conditions [97,98]. 

### 3.12. Transition to Adult Care

Transitioning pediatric and adolescent care to Adult Care is challenging for any child, including many children with RD [100]. The change in family support that comes with increased independence from caregivers to self-management of one’s own care brings many challenges and needs. In the past, individuals with childhood onset RD may have had a poor prognosis and may or may not have lived into adulthood. With advances in medicine, these children are living longer and adult healthcare teams to whom they turn may be even less familiar with the childhood onset RD than those with pediatric training. Additionally, expectations may be different in adult settings. For some children, chronological adulthood does not mean caregiving needs are gone. When some individuals with RD become legal adults, they may need various levels of ongoing assistance. Some children with RD have intellectual disabilities or differences and find it difficult to negotiate the expectation of being independent and capable of their own decision-making without someone else present. Additionally, for children who have neurodevelopmental delays, supportive services such as those of Child Life Specialists are not typically available in adult hospital or clinic settings. Child Life Specialists are trained in promoting developmentally appropriate coping skills to minimize adverse effects of stressors related to healthcare encounters and procedures [101]. In some countries, legal structures require that youth with RD be evaluated or determined to require a guardian to manage certain aspects of their livelihood. This may be required to access funds allotted for their housing or care in adulthood. Considerations may include medical decision making, financial holdings, housing, legal rights, navigation of healthcare systems, and optimizing independent functioning and safety as possible given their needs [102]. There are many complexities and hurdles to determining if assistance needed and legal and insurance changes to navigate. Additionally, for individuals with RD and physical differences or disabilities, accessible transportation, services, and healthcare offices may be difficult to negotiate independently. 

## 4. Recommendations

The material presented here represents an overview of the many facets of caring for children with RD that impact their psychosocial wellbeing. While some evidence exists for improving systems of care and care delivery, much remains to be done to advance the state of supports across settings for children with RD and their families. In what follows, specific pathways are proposed to achieve high priority recommendations that may further herald this call to action for leaders in healthcare, education, research, and policy. 

### 4.1. Support Pathways

#### 4.1.1. The Child and Family

In Germany, CAREFAMNET examined the many gaps between medical and psychosocial health for children with RD and their families [103]. They found that psychosocial care is not standard part of routine care for these children and families. They highlight a need for improvement to facilitate access to psychosocial care and support, expand services to all family members, strength, and expert patient organizations, simplify application procedures and more cooperation between funding agencies, strengthen low threshold services, integrate psychosocial care, and promote interdisciplinary collaboration and networking. Many families of children with RD face substantial burdens related to the time and intensity of daily cares, frequent tradeoffs to balance caregiving with employment and other family needs, the social isolation of their unique caregiving experience, and navigating a complex and often fragmented healthcare system. Improving caregiver supports can help decrease caregiver burden and help families connect with peers [40]. Many communities have peer support networks both for parent caregivers and for children with RD themselves. Such networks may allow families with a child having an RD to connect with another family of a child with the same condition. If that is not possible, connecting with a family with similar lived experiences of RD, such as the challenges of a diagnostic odyssey or daily life with medical technology dependence, can offer valuable support. Diagnosis-specific national organizations and support groups are also a resource for caregivers to connect with other families affected by an increasing number of rare diseases. In the USA, the National Organization for Rare Disorders (NORD) has established a database of these organizations on their website [104]. There are also organizations that provide support and resources based on a child’s developmental or medical needs rather than a specific diagnosis. Clinicians should help families connect with community organizations that support educational and community advocacy efforts on behalf of children with complex medical needs. 

An additional support for families may come through connecting them with community organizations that support educational and community advocacy efforts on behalf of children with an RD or complex medical needs. Such support groups may take the form of being informational, resource-oriented, providing a peer mentor, or family-to-family support. Parents are recommended to seek support with others who have shared similar lived experiences although some RD specific connections may be available. One such parent support program with global accessibility and impact includes Parent to Parent online support groups that are arranged by country [105]. In New Zealand, for example, Parent to Parent has support groups in twelve different locations [106]. These resources aim to connect parent caregivers across the globe with other parents who understand their unique circumstance. Parents may choose to match with another parent, for example, by disease, location, or special healthcare need [107]. In-person community and social connection may be built by attending informational, disease specific group events, or supportive gatherings. One example in the USA is Hope Kids which provides ongoing events, activities and a powerful, unique support community for families who have a child with a life-threatening medical condition [108]. The mission is to surround these remarkable children and their families with the message that hope is a powerful medicine. 

#### 4.1.2. The Siblings

The presence of a family member with RD provides opportunities for increased empathy, responsibility, adaptability, problem solving and creativity. Siblings themselves highlight that obtaining support from friends, peers, and support groups were essential as a positive force in managing changes in their lives, while negative reactions from others was a detriment. What mediated this was coping, acceptance and adjustment [82]. They note that families and healthcare teams may underestimate the emotional responses and needs of siblings due to their presentation of self-sufficiency and adaptation to additional responsibilities despite experiencing elevated levels of stress. Such evidence about the importance of addressing sibling’s psychosocial needs has led sibling interventions to be incorporated as a standard of care in pediatric oncology, for example. Thus, this is a great opportunity for parents and healthcare providers to meet siblings’ needs in numerous ways. 

Parents, be aware that while attending to the needs of the child with RD, you may be neglecting—or creating unfair expectations for—your other children.Siblings can learn to participate in the family and feel pride and love in helping their brother or sister with their health.Try to establish some balance between the needs of your child with a chronic health problem or disability and those of your other children.Keep in mind that siblings need to have honest information about the condition and to have their questions listened to and answered.Spending small amounts of quality time with each child individually as much as possible may help.Support groups involving other siblings in a comparable situation can play a pivotal role in siblings’ coping and thriving amidst this challenging situation.

One example of a resource for sibling support that has a global impact includes the Sibling Support Project and Sibshops including SibTeen, Sib20, and SibNet [109]. These are online communities for siblings across various age groups, which allow thousands of siblings of youth with complex medical conditions from around the world to connect with their peers to both receive and provide much needed support [109] and can be accessed here: (https://siblingsupport.org accessed on 4 December 2021). 

### 4.2. Support and Collabortion Pathways

#### 4.2.1. Behavioral Health

Supportive care coordination and shared care plans have been associated with improved parental mental health for families of CSHCN [63,64]. To address behavioral health needs in the settings where children with RD and their families are found, certain adaptations to an integrated care model such as the Medical Home, or specialty care center, may prove useful. Integrated care embeds behavioral health providers such as psychologists and social workers and counselors into the healthcare setting. Visits to the Medical Home provider may yield a consult with an embedded mental health provider as a first point of contact for an emotional, behavioral, or psychosocial need, providing immediate access to care [110]. To optimize family wellbeing, it is important that psychosocial care be needs-oriented for children and their families rather than diagnosis-oriented [103]. Families may prefer not to add another care provider to their family’s team nor to add more appointments on the family calendar. However, there are situations such as depression with suicidal ideation or reduced effectiveness in completing demands due to depression, anxiety impacting sleep and daily effectiveness, sleep disorders reducing one’s restoration that could sustain caregiving and wellbeing. These are examples when evidence based behavioral healthcare is recommended and can be positively impactful for the parent caregiver and the child patient [111]. Services for adult caregivers of a child with RD may be identified through your healthcare system that cares for your child. Families are encouraged to ask the provider for the child with RD for vetted referrals who understand the unique circumstances of caregivers of a child with RD. Telehealth counseling or psychotherapy may be available that would be otherwise not accessible due to location or transportation and other time- and- effort- based barriers. 

#### 4.2.2. Partnership 

The overarching model that best accommodates psychosocial needs of children with RD and their families is the PFCC approach because it is known to improve patient and family satisfaction, patient self-management, and physical and mental health outcomes. The foundational principles of patient- and family-centered care include valuing patients and families as members of the healthcare team, ensuring inclusive communication, and harnessing technology to promote access to health information [112]. 

### 4.3. Patient Family Advisory Councils or Boards

Family advisory boards, councils, and volunteer service opportunities that provide shared experiences and social support, sense of community. Importantly, for patient family engagement to be effective in partnering in the healthcare of RD, those serving on the boards must reflect the views, needs, and culture of those they represent. In one large pediatric health system in the USA, barriers to participation in PFACs were found to vary by race/ethnicity and socioeconomic status, with those from Black or less affluent families underrepresented in these boards [113]. If a sizable portion of those served by the health system are of a particular racial or ethnic group, or spoken language, their SDH should be addressed by the institutional policies advocated for by the representatives of the current board. Richard and colleagues outlined recommendations for developing, implementing, and sustaining PFACs [114]. These included four themes and are summarized as follows: Use evolving recruitment methodsPrepare for effective participationEnsure diversity within PFACsOutline terms for orientation and participation.

PFACs should recruit in an ongoing manner to maintain adequate participant numbers in light of the ongoing changes in circumstances and needs of families. Important to acknowledge is that members may agree to serve a particular term (e.g., one year) with the option to renew their commitment annually, but that they should feel free to withdraw their participation at any time should their needs change. PFACs should produce “living” reference documents that establish regulations around membership and recruitment that provide guidelines to PFAC members, including how to recruit and retain members and outline how they would be involved in the council. Training was completed using an orientation manual that includes expectations, ways to stay engaged and connected, legislative requirements such as accessibility and privacy as well as in-person orientation on how they can tell their story, teaching them about the organization and what they can expect. Members should be the ones who choose when the meeting dates are going to be, how they want to set up the council. In a pediatric setting diverse representation would include the pediatric voice as well as various ethnic, religious, geographic, and socioeconomic backgrounds [114]. It was stated that the PFAC should look like your waiting room. Some strategies utilized to promote diversity within the PFAC include targeted recruitment for members of underrepresented groups; recruitment of individuals who work with members of minority and underrepresented groups [113]. 

Next, for the PFAC to thrive, the authors recommended the PFAC leaders to address logistics to promote attendance, for example, virtually or in person. On an ongoing basis, address open communication and all barriers to participation among members. This may be accomplished by setting a standard for continuous communication that is established from the outset of the PFAC. Provision of certified interpreter services that allow individuals with language barriers to participate effectively and efficiently; virtual participation options for those who may not always be able to physically attend meetings; and finally, meeting with individual groups outside the institution to gather information from minority groups that can be brought back to the PFAC. One recommendation is to consider compensation of parents who donate their time and expertise to PFACs rather than considering them vetted volunteers for the health system. This may improve the ability for parents who have not participated due to financial barriers or related circumstances to participate through financial support for their time. This could enhance diversity and inclusion and representation of all families within these advisory boards which is so important to guide health systems in serving diverse populations. 

## 5. Calls to Action

Healthcare Teams.

### 5.1. Include the Child Directly

Families and healthcare teams can positively influence the child’s acute care experience by intentionally including the child in decision making processes while admitted to the hospital and this is expected to increase coping through improved engagement and control with their own outcomes [25]. Although there is limited research on communication between healthcare staff and children with communication barriers or who are nonverbal, parents have recommended that we speak directly to their child rather than rely on the parent to serve as a go-between. In fact, parents report feeling more anxious if they are the only communicator for their child and want health team to make more of an effort to involve the child in their care. It is known that children feel and need competence in medical visits and being a part of medical decision -making about their lives is empowering. Specifically, parents ask the care staff to show the child what to expect and help them feel in control using communication at their level. Parents ask that staff learn about the child’s unique way of communicating and learn how to use it. For example, if a child uses an adaptive communication device, sign language, or a communication board be prepared so that the child can interact in the visit. Communicating with the child can increase their sense of safety and security. Finally, parents ask that if one member of the healthcare team learns about ways to communicate effectively with the child, share this with the rest of the team so that the family does not have to “start over” in retelling their story with each visit [115]. See Figure 3. 

### 5.2. Leverage Collaborative Communication

Taking time to hear the child and family’s concerns and to partner with them in their care is the aim of care conferences. These are meetings with the parent and medical specialists to discuss the patient’s care, express concerns, and share information. Care Conferences are often non-billable and thus can be rife with limitations in the medical system driven by revenue productivity. However, care conferences add value to the collaboration between patient, family and healthcare team that cannot be built elsewhere through other methods of consultation among providers or in office visits. The electronic health record (EHR) provides a means for messaging to facilitate information sharing and communication with families. This allows for documentation to be saved and referenced later by the family and the care team. Families are recommended to keep a journal or binder of information with questions, answers, and concerns to bring up in visits. This allows the parent and the care team to ensure that all questions are addressed, and that solutions and interventions are tracked over time. Parents are encouraged to ask for Infographics and visual aids for difficult to understand routines, cares, or medication regimens, mobile apps and pill minders for timed dosing and improved adherence to prescription medication plans. Parents are recommended to offer their assistance to the care team as an observer or data collector in the home to inform medical or behavioral health decision making. Behavioral health providers may provide charts or checklists to help parents or school staff track and share with the care team any clinically relevant home behaviors and symptoms. 

Care coordination for CMC with RD may improve parental mental health [63,116], in addition to the other expected benefits for the child. While not all healthcare providers will have the same resources, healthcare teams must proactively communicate to ensure coordinated care. There are tools to help with care coordination. Proactive guidance when available such as use of emergency information forms, action plans, comprehensive care plans can be useful in helping families to communicate essential information. Families and individuals with RD are part of the care team and should participate in creation and updates of care plans. Additionally, inclusion of the parent voice in clinical practices and improvement work is key to optimizing care for children with RD. Healthcare teams are encouraged to approach the provider-patient/family relationship with curiosity and an open mind, and frame caregiver expertise as an asset to that partnership. 

Care mapping is a family-driven, person-centered process to highlight a family’s strengths and communicate both the big picture and small details of all the resources needed to support a child and family in a snapshot. The process of care mapping has served as a useful tool for care coordination and patient and family engagement [117]. See Figure 4 for links to instructions and additional resources to engage with families in care mapping. 

A care map depicts that the web families navigate is not limited to medical providers alone. The healthcare team may include professionals from many fields such as a pharmacist, social worker, psychologist, dietician, nurse, care coordinator, patient navigator, therapist, genetic counselor, or a medical librarian. The child and family are also an essential and integral and central member of the care team. Regular communication between care teams, outpatient and inpatient and use of care co-management guidelines or customized care plans can enhance care coordination and ensure healthcare teams are on the same page. Medical care is only one aspect of the experience of a child with RD and coordination with community and educational services is also essential. An integrated care model is recommended, when possible, as it may increase behavioral healthcare access (embedding behavioral health and psychosocial supports within the medical setting and team). Care teams can collaborate with a patient and their parents and the child’s school and home- and community-based care providers to develop a shared plan of care that reflects the many settings that contribute to a child’s care. Partner with schools and other agencies working with the family to glean data points to guide decision making about interventions such as medication dosing, behavioral interventions. Examples of care coordination tools can be found here: https://www.ahrq.gov/ncepcr/care/coordination.html (accessed on 11 March 2022) [118].

*1.* 
*Facilitate Transition to Adult Care*


To improve transition to adult care for individuals with RD, healthcare teams should discuss transition early starting in adolescence. Teams can discuss ways to prepare individuals with RD and their families for adult healthcare and system navigation. These may include initiating testing if it is unclear if an individual with RD will need any level of legal guardianship or other supports and discussing the process. Healthcare teams can aid by creating a transition summary and communicating with potential or new adult healthcare teams during the transition period [119]. Healthcare systems must recognize this challenge as well to better support individuals with RD and provide resources but also adequate training for adult physicians [120]. Providing a transition coordinator is one intervention that may be useful for facilitating a smooth transition. The ‘why’ behind transition cannot be understated; when young adults with RD reach the age of transition to adult care, this opens space for new young children with RD and their families to access high quality care [102]. 

*2.* 
*Access and Share Essential Information*


Individuals with RD and their families often have a long quest for a diagnosis or understanding the RD. Healthcare teams are encouraged to partner with families and to encourage the search for answers. One mechanism of action is participation in clinical trials and in helping families to connect to case reports in the medical literature that will provide added information about their child’s RD. Teams can encourage and facilitate connection with other families or networks to share and learn. Families and healthcare teams are also encouraged to discuss where to find reliable sources of information. This may range from online sources, but also expert medical consultations or second opinions, clinical trial opportunities, and expertise from other families with lived experience. Families should be encouraged to bring questions to their visits, including any information found online to discuss. Utilizing and providing access to a medical librarian may also be critical in helping to find accurate information for both providers and individuals with RD and their family. Families are encouraged to ask their care providers what reputable sources or societies are disseminating innovative, up to date, evidence-based or vetted information about their child’s condition. From this starting point, families may find they are able to determine sources they can trust and references they can rely upon. Families are encouraged to ask questions on what is known and what is being explored in clinical trials, for example, but not know how much weight and hope to place on what may or may not be possible.

To address challenges related to accessing available information and care instructions, healthcare teams should use health-literacy-informed strategies. Healthy people 2030 recognizes that it is not just the individual that determines health literacy, but it is also affected by the degree to which organizations equitably enable individuals to find, understand and use information and services [79]. Many standard patient education child health handouts may not always be applicable or sensitive to the needs of individuals with RD and must be further personalized. For instance, individuals with RD may have complicated medication regimens or care routines or dietary needs. Additionally, an individual’s health literacy can worsen during times of stress [81]. It is known that individuals with RD may have complicated care instructions and their parents may experience elevated levels of stress. To promote understanding of instructions, the use of plain language with primary-grade-level readability (sixth grade or lower in the USA system), and visual graphics when possible, are recommended [121]. The “teach back’ method can be used to check for understanding by asking children and parents to teach back what they heard in their own words. Although this technique can cause some discomfort at first, with practice, it is effective to increase patient-centered communication and effective engagement [122]. Healthcare teams can mitigate disparities associated with low health literacy by using health-literacy-informed strategies including limiting information, action-oriented instructions, plan language, demonstration, teach back, supplementing verbal with written information and pictographic and multimedia materials [121]. Additionally, for those individuals or families that do not share a common language with the healthcare team, having interpreter services and translation for written materials is essential for communication.

Health literacy is one social determinant of which there is no need to screen for, as all individuals benefit from clear communication. However, healthcare teams should screen for the impact of SDH. While this is a recommended practice for all pediatric patients, there is a particularly high likelihood of SDH factors impacting health outcomes for children with rare diseases, given their likelihood of having complex chronic medical needs. The complex financial burdens these families experience should be considered, and families should be helped to connect with financial supports and assistance programs for which they qualify [123]. Integration of social work into the healthcare team can be extremely beneficial to families of children with RD, particularly when it is structured as longitudinal case management to address these families’ dynamic needs over time. 

*3.* 
*Sustain Strategically*


Provider wellbeing is a vital consideration in sustaining the medical provider role alongside a child with RD and their families’ difficult journey. While provider wellbeing initiatives are emerging within healthcare globally, recommendations for medical providers to bolster their longevity and wellbeing and reduce ‘burnout’ are a key consideration when caring for the child with RD. Turnover of health professionals has also been identified as a concern for those with RD. Families have identified as a stressor or dissatisfier the need to continually update new providers. Health systems are recommended to utilize collaborative care teams to sustain each other more than sole providers. Care teams may hire professional roles for time-sensitive and time-intensive tasks. For example, a clinical pharmacist can supplement the care team for children with RD to manage polypharmacy, medication reconciliations and this has been shown to improve provider burnout [119]. They may also partner with schools and other agencies working with the family to glean data points to guide decision making about interventions such as medication dosing and behavioral interventions. 

Improving electronic health record functionality with documentation, correspondence, consultation, and communication among team members, and reduce time-wasters and duplication of efforts with processes can improve provider wellbeing [124,125]. In fact, simply using messaging within the EHR for consultation reduces provider burnout [124,125]. Utilize the team’s expertise rather than trying to do it all yourself. Plan joint or cascading visits so that the family may come to clinic for a visit but see multiple professionals either in person or virtually in the context of that visit. Engage with families as partners through efficient communication, for example, using modalities that double as documentation such as patient portal that populates into the electronic health record [126]. Honor the expert medical parent with partnership to serve as your eyes and hands in the home setting toto increase efficiency in your outpatient diagnostic workup and treatment planning [38].

### 5.3. Educators

Schools are called to provide inclusive education through intentional collaboration with families and healthcare teams. When teachers and students understand the child’s experience and acknowledge their disease, a student may experience a reduction in “feeling different,” an increase in their emotional wellbeing, and a reduction in caregiver worry [92]. Schools have the opportunity to provide a safe, accepting, and inclusive environment for children and may improve quality of life [93]. To reduce disadvantages in their education process and to promote a supportive environment, it is recommended that schools increase their awareness of the experiences of children with RD and take actionable steps to make the school environment most accessible [92,93]. Healthcare teams have an opportunity to promote communication and shared understanding between the settings where the child spends their time. Additionally, healthcare teams are recommended to ask families for their permission (release of information) to communicate with the school and enhance the connections between the school and healthcare setting to coordinate needs for the child.

### 5.4. Investigators

At this time, three areas of advancement in research have been identified. First, research to date focuses on the experiences of caregivers, but the voices and experiences of the child with RD are not quantified or described. This is concerning as many of the psychosocial and medical treatments are focused on improving the child or adolescents’ wellbeing, but there are scant data guiding clinicians on best practices that are congruent and affirming based on these youths’ experiences. Thus, a first step is to direct the investigator’s attention to exploring and describing the experiences of youth with RD and to include them in the research process. A model that might be beneficial to apply is community-based participatory research, where youth are guiding the research developmental process. Researchers may serve as a vehicle to elevate caregiver voices in these spaces in appropriate and effective collaboration. Second, there is a need for translational research, where studies highlight whom interventions will benefit and under what specific conditions. While correlational studies are necessary, at this time there is a lack of clarity and direction in how these research findings may best be applied in clinical settings. Third, there is a lack of intersectionality in understanding youth with RD, where there is limited knowledge on race, ethnicity, gender, SES, geographic location, and access to specific resources. Within the context of the USA, the social construct of race has been the main driver in inequities in service access. Studies that lack this intersectional lens are doing a disservice to a large section of the population and are further perpetuating inequities. Thus, investigators are encouraged to use an intersectional framework in their research design, while using a wholistic rather than additive approach. 

### 5.5. Policy Advocates and Change-Makers

Investigators may collaborate with caregivers to not only examine their experiences but to also elevate their voices in policy spaces. Policy change relies on advocacy by families of RD, who are already stretched thin by constant caregiving responsibilities. Expecting these families to speak up to promote change represents a major barrier to progress addressing the policy needs of this population. Policy change requires the perspectives of caregiver and patient advocates to highlight and address their needs. However, caregivers of youth with RD are often prevented from participation due to practical barriers, such as providing round-the-clock care. This is a major barrier that prevents progress in policy that is reflective of this population’s needs. Policy-makers are urged to act on the significant gaps in equity and provision for this population with specific recommendations highlighted here. 

#### 5.5.1. Improve SDH at Multiple Levels

SDH are defined by the World Health Organization as the conditions in which people are born, grow, live, work and age [127]. When considering the modifiable contributors to an individual’s health outcomes, medical care is estimated to account for ten to twenty percent [128], with social determinants of health dwarfing the impact of direct clinical services. Interventions to address SDH should be focused on the community and population levels in addition to individual screening, in order to systematically address these common needs, which have a disproportionate effect on families of children with chronic medical needs [129]. Connecting families with care coordination services either within the medical home or in partnership with a community organization may be a successful means of addressing SDH which may also improve the family’s experience of other aspects of their child’s care [130]. The provision of care coordination services is often limited by poor reimbursement within established payment models for pediatric medical services, making it financially unsustainable for centers to offer. Recognize the impact of caregiver-provided medical care for children with RD, and advocate for enrollment in home nursing care and patient care aide services when medically appropriate [38,45,131]. Augmenting existing funding streams and creating new financial incentives for care coordination has the potential to improve access to this important service for families of children with RD [130]. 

#### 5.5.2. Improve Home and Community Based Services

Improve funding for home-based services including private duty nursing, skilled nursing visits, and patient care aide services. The authors are writing from the perspective of the USA where there is a long-standing shortage of pediatric home nursing services. Improved funding and support for these services would allow children to receive care in the least restrictive environment possible, decrease preventable hospitalizations [131] and contribute to family wellbeing by decreasing parents’ need to improvise this skilled caregiving on their own. While other countries’ statuses may differ regarding this service, this USA perspective highlights the importance of bolstering the fragile network of home- and community-based services on which these families rely. It is recommended that policy-makers invest in robust paid family leave policies, childcare programs equipped to serve CSHCN with RD, and paid family caregiving, which can mitigate the impact of children’s chronic health needs on families [44]. Additionally, advocating for the establishment and expansion of these services and local, regional, and national levels is necessary. 

#### 5.5.3. Improve Accessibility in Communities

It is recommended to improve access to wheelchair-accessible transportation and accessibility of community spaces and activities to allow children with RD to participate fully in family and community activities regardless of disability status [85,86]. As many existing legal statutes are not sufficient for ensuring this accessibility, policy-makers should consider bolstering this legislation to improve its effectiveness and provide resources for communities and organizations to adapt their spaces and programs more successfully to be inclusive. 

## 6. Limitations

There are limitations to account for when considering the results of this review. First, our sampling of the literature may have missed articles not included in English or within the databases used. Second, our recommendations may not represent consensus among all healthcare professionals caring for this population and may reflect the sampling of the available literature. Third, the authors all practice in the USA which may bias or slant our approach to the topic influencing how recommendations are framed. Finally, very little research specifically addresses the psychosocial needs of children with rare diseases. Therefore, much of the current knowledge is gleaned from a sampling of studies that address one or more facets of either a subset of the population or more global the psychosocial concerns. There is much to be learned from the growing body of literature regarding relevant intersectionality of multiple identities represented by children with RD, including children with medical complexity or disabled children, and these populations’ needs may not capture those of children with RD. 

## 7. Conclusions

This review has endeavored to provide a thorough examination of psychosocial considerations for the child with RD. In doing so, several gaps in the medical literature to bolster evidence-based care, psychosocial support, and access to resources in society at large for children with RD have been identified. Therefore, existing information on the psychosocial considerations of populations of children with chronic conditions, including CSHCN, CMC, and what little is known about RD specifically, is examined. In summarizing literature through a purposeful sampling of the scientific literature, recommendations to integrate what is known of current best practices, accessing optimal resources, and proposals framed as ‘calls to action’ are given to elevate the quality of life and promote evidence-based care for children with RD, their families, and healthcare teams. Understanding the psychosocial considerations for children with RD will hopefully energize future endeavors to better understand and address the needs of these remarkable children and their families.

## Figures and Tables

**Figure 1 children-09-00933-f001:**
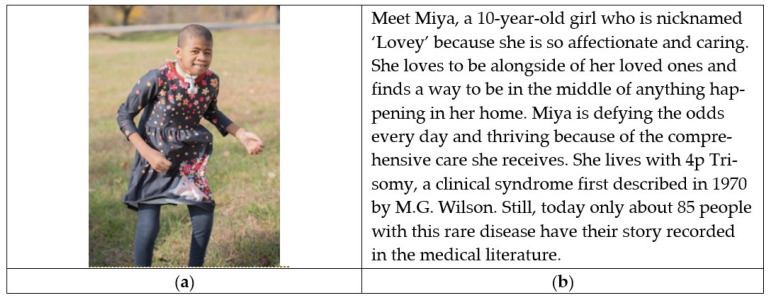
(**a**) Photo Captions: #RareDisease #RareIsntRare; (**b**) Miya’s Story [14].

**Figure 2 children-09-00933-f002:**
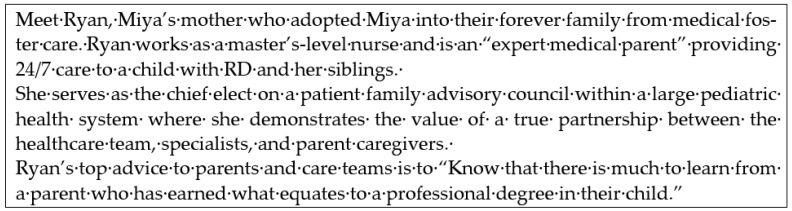
The Expert Medical Parent [14].

**Figure 3 children-09-00933-f003:**
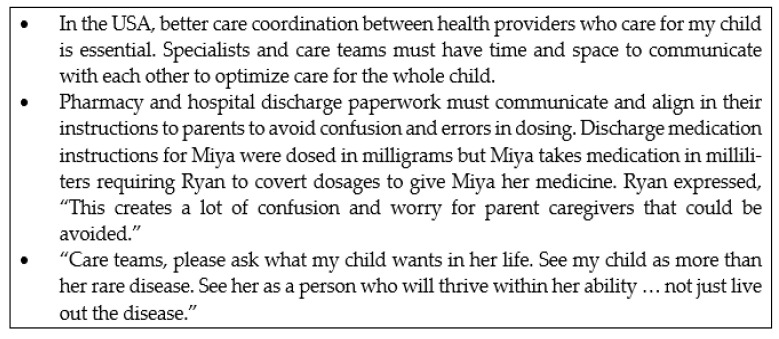
A Parent Voice: Calls to action that Ryan values most for Miya [14].

**Figure 4 children-09-00933-f004:**
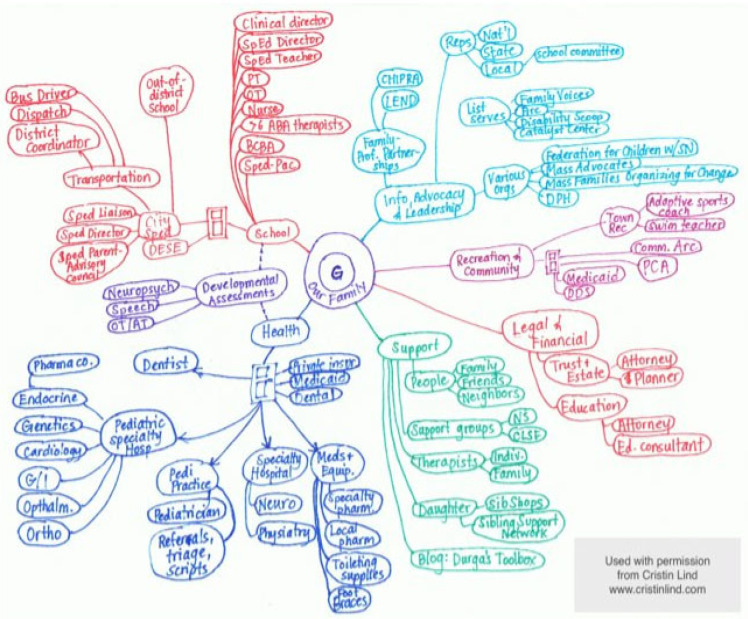
Care map created by the mother of a child with medical complexity to pictorially represent aspects of the care her son requires to be coordinated. Used with permission from Cristin Lind, http://www.childrenshospital.org/integrated-care-program/care-mapping accessed on 4 December 2021.

## Data Availability

Not applicable.

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
