# Peer review of "Psychosocial Considerations for the Child with Rare Disease: A Review with Recommendations and Calls to Action"

_children, 2022, doi:10.3390/children9070933_

Round 1

Reviewer 1 Report

A very interesting paper about one actual topic with a lot of potential. Anyway, there are few comments about the article I would like to make in order to improve its publication:

- Abstract doesn´t mention the methodology of the study, at less not clearly and briefly explained. I highly recommend to use the sequence Intr/Method/Results/Conclusions also in the abstract.

- As a general comment, I highly recommend to write the paper in third person (I know there are differences about this topic, but for the scientific sound it´s clearer to avoid third person excepting in discussion and conclusions.

- The structure of the paper is really confusing. There is one Introduction, one Discussion and Recommendations and Conclusions. It´s really difficult to make a mental structure about the paper in order to connect one topic with others.

- Related with the previous, the psychosocial factors and why they were selected are not really clear in the paper.

- There is no Methodology description, there are revisions of other studies, so it´s difficult to support "We have demonstrated the imperative to advance the state of supports across settings..." (line 481)

- In conclusions part, line 914, the authors express "In this review...". It can be useful if the authors clarify how has been done the review, why the practices included in the text have been selected, if there are criteria for this selection...

- Maybe a part abut biases and/or limitations of the review can be useful for better understanding of the paper

Author Response

Response to Reviewer 1

Thank you so much for your positive and constructive feedback about our review article entitled, Psychosocial Considerations for the Child with Rare Disease: A Review with Recommendations and Calls to Action. We especially appreciate your positive feedback about the topic having a lot of potential. Please see the attached revised manuscript with tracked changes. Also, please see below the detailed revisions made at your recommendation:

Lines 25-27 of the Abstract describe the methodology of the study specifically as you recommended. At your recommendation we clarified sequentially in the Abstract those sections for the abstract that mirror sections in the body of the review with key words to clarify flow.

In response to your recommendation about use of third person voice throughout the review, we have modified throughout the entire manuscript to reflect consistent use of third person. Please see the attached revised manuscript with this modification.

Thank you for your constructive feedback about the structure of the manuscript. Please see the attached manuscript revisions that include the following at your recommendation: 

In lines 496 i.e., “Recommendations” and 671, i.e., “Calls to Action” we revised those headings to more accurately describe the section contents, and we deleted the term “Discussion” to enhance clarity of the structure.

Based upon your final suggestion that is “useful for better understanding of the paper,” we added a “Limitations” section heading on line 940 with that section noting any potential biases and a “Conclusions” heading on line 954 with that section clarified in lines 955-969.  

We added the Materials and Methods section on lines 88-97 and separated this from the Results section heading on line 99 which more accurately describes the contents of this review article. The addition of the Methods section with detailed description of our purposeful sampling of the scientific literature, inclusion and exclusion criteria of articles reviewed is now revised to addresses your feedback about “the psychosocial factors and why they were selected are not really clear in the paper.”  

Lines 499-502 have been edited not only for voice but specifically to address your comment: “It is difficult to support the assertion, ‘We have demonstrated the imperative to advance the state of supports across settings…’”

Line 962 of the “Conclusions” section has been revised according to your recommendation to elucidate how the review was conducted and why specific topics were included or excluded based upon the sampling of the literature as outlined in the Materials and Methods sections, as mentioned above.

Again, thank you so much for your thorough review of the review manuscript.

Reviewer 2 Report

The paper addresses, from my point of view, an interesting and topical issue such as the study of psychosocial factors involved in children with rare diseases.

I think the presentation of the study is very good, as well as the approach to possible recommendations for intervention with this population. However, there is one section that I did not find in the review and that I think is very important, I am talking about the methodology used in the literature review. Without this section, it is difficult to replicate this study. Furthermore, without this important section, I consider this work to be more characteristic of a publication in a book as a chapter than in a scientific journal. In one of the sections it talks about discussion and I believe that there is no discussion, since what it describes are the recommendations for possible interventions. I insist that the way the study is written, it would be more appropriate to publish it in a book chapter than in a scientific article.

Author Response

Response to Reviewer 2

Thank you so much for your positive and constructive feedback about our review article entitled, Psychosocial Considerations for the Child with Rare Disease: A Review with Recommendations and Calls to Action. We especially appreciate your positive feedback about the presentation and recommendations for intervention with this population. Please see the attached revised manuscript with tracked changes. Also, please see below the detailed revisions made at your recommendation:

Note that on page 3, line 86 the Materials and Methods section has been added and fleshed out in detail, with appropriate modification of headings. As you indicated, the addition of this section will importantly allow for the replication of the study. Additionally, this section will clarify for the reader the structure of the review article as it flows easily into the Results section in which the review of the topic is elucidated.

We also appreciate your observation about the format of the manuscript. In response, we revised and clarified these specific sections and flow and thus, we put forth this revised manuscript as a review article in this special issue of Children for Psychosocial Considerations for Children and Adolescents Living with Rare Diseases, instead of as a book chapter.

Thank you for bringing to our attention your concern with the discussion section heading as not fitting the content of the section to follow. Please see the revised headings on lines 496 i.e., “Recommendations” and 671, i.e., “Calls to Action” in which we have revised those headings to more accurately describe the section contents, and in which we deleted the term “Discussion” at your recommendation.

Again, thank you for your review.

Round 2

Reviewer 1 Report

I really appreciate the efforts of the authors for the improvement of the paper and the inclussion of all comments and proposals

Author Response

Thank you very much for your assistance in reviewing this review paper. We appreciate your input. 

Reviewer 2 Report

I consider that with the changes introduced the article has highly improvement.

Author Response

Thank you so much for your input on this review article and for your reviews. We appreciate your assistance in improving this for publication.